# Regulation of Meat Duck Activeness through Photoperiod Based on Deep Learning

**DOI:** 10.3390/ani13223520

**Published:** 2023-11-14

**Authors:** Enze Duan, Guofeng Han, Shida Zhao, Yiheng Ma, Yingchun Lv, Zongchun Bai

**Affiliations:** 1Agricultural Facilities and Equipment Research Institute, Jiangsu Academy of Agriculture Science, Nanjing 210014, China; 20220049@jaas.ac.cn (E.D.); 20210044@jaas.ac.cn (Y.M.);; 2Key Laboratory of Protected Agriculture Engineering in the Middle and Lower Reaches of Yangtze River, Ministry of Agriculture and Rural Affairs, Nanjing 210014, China

**Keywords:** deep learning, MOT, YOLOv8, duck activeness, photoperiod, animal monitoring, breeding welfare

## Abstract

**Simple Summary:**

In commercial duck farming, the photoperiod is a crucial farming parameter that can influence physiological and health indicators. Among these indicators, duck activeness, which is directly affected by the photoperiod, is one of the most intuitive and currently a hot research topic in the field of animal welfare. However, there are existing limitations in the calculation methods for activeness, which often focus on relationships between adjacent frames, neglecting medium- to long-term features and sometimes lacking intuitiveness. Additionally, the relationship between duck activeness and the photoperiod has not been clearly defined, making it challenging to provide practical guidance for production. This study introduces a duck activeness estimation method based on machine vision technology. It involves tracking the movement of group-raised ducks over a 6 min period. The average displacement of all ducks within each frame is used as an indicator to measure activeness, resulting in more accurate results. This method is applied to assess duck activeness under different photoperiods, and the experimental results are analyzed to determine the most suitable lighting cycle for duck farming, demonstrating the superiority of the model. The proposed method and the conclusions regarding the optimal photoperiod provide both a methodology and data support to enhance duck breeding and farming efficiency.

**Abstract:**

The regulation of duck physiology and behavior through the photoperiod holds significant importance for enhancing poultry farming efficiency. To clarify the impact of the photoperiod on group-raised duck activeness and quantify duck activeness, this study proposes a method that employs a multi-object tracking model to calculate group-raised duck activeness. Then, duck farming experiments were designed with varying photoperiods as gradients to assess this impact. The constructed multi-object tracking model for group-raised ducks was based on YOLOv8. The C2f-Faster-EMA module, which combines C2f-Faster with the EMA attention mechanism, was used to improve the object recognition performance of YOLOv8. Furthermore, an analysis of the tracking performance of Bot-SORT, ByteTrack, and DeepSORT algorithms on small-sized duck targets was conducted. Building upon this foundation, the duck instances in the images were segmented to calculate the distance traveled by individual ducks, while the centroid of the duck mask was used in place of the mask regression box’s center point. The single-frame average displacement of group-raised ducks was utilized as an intuitive indicator of their activeness. Farming experiments were conducted with varying photoperiods (24L:0D, 16L:8D, and 12L:12D), and the constructed model was used to calculate the activeness of group-raised ducks. The results demonstrated that the YOLOv8x-C2f-Faster-EMA model achieved an object recognition accuracy (mAP@50-95) of 97.9%. The improved YOLOv8 + Bot-SORT model achieved a multi-object tracking accuracy of 85.1%. When the photoperiod was set to 12L:12D, duck activeness was slightly lower than that of the commercial farming’s 24L:0D lighting scheme, but duck performance was better. The methods and conclusions presented in this study can provide theoretical support for the welfare assessment of meat duck farming and photoperiod regulation strategies in farming.

## 1. Introduction

The photoperiod is one of the crucial environmental factors in poultry production [1] and can regulate biological and physiological functions by adjusting the circadian rhythms of poultry [2,3,4]. Birds are photosensitive animals, and altering the photoperiod can impact their growth and production performance [5,6] as well as animal welfare and health [7]. Maintaining an appropriate photoperiod in commercial farming is vital for ensuring stable production and safeguarding profits.

Currently, in China, continuous or near-continuous 24 h lighting schemes are implemented in meat duck production to maximize growth rates and feed intake for improved production performance [8]. However, continuous lighting can lead to metabolic syndrome and compromise the welfare of meat ducks [9]. Some researchers and farms have started investigating the optimal photoperiod for meat ducks, but existing studies mostly focus on post-slaughter performance metrics such as growth performance, carcass characteristics, meat quality, physiological stress (hormone levels), and blood parameters [10,11]. There has been limited attention given to the physiological behavior of meat ducks. During the farming process, poultry activeness serves as a crucial reference for assessing poultry welfare and calculating poultry gait scores [12]. Among the various physiological behaviors that poultry can exhibit, poultry activeness is the most intuitive and prominent. Using poultry activeness as an indicator to evaluate the impact of the photoperiod on duck physiological behavior is the most convenient and practical approach.

Methods for calculating poultry activity include manual scoring of animal behavior or fitting scores through image processing. The former involves the analysis and interpretation of animal-based information by human experts, which can be challenging, time consuming, and costly. The latter enables automated information processing. Leroy et al. [13] established a computer vision system to study the behavior of caged hens, including standing, walking, and pecking. Dawkin et al. [14] used video footage of broilers in commercial housing and optical flow analysis of group movement from closed-circuit television to measure the activity of broilers and further assess their welfare. Lao et al. [15] defined a pixel set in the current image that belonged to the region of chickens but did not belong to the region of chickens in the previous frame. Based on this, they calculated the horizontal activity and vertical activity of the chickens, thereby assessing the chickens’ activity levels at different time intervals. Yang et al. [16] proposed a method in their study evaluating the impact of an elevated platform and a robotic vehicle on chicken welfare. This method involved calculating the ratio of motion pixels of birds in consecutive images to the total representative pixels of the entire flock (total white pixels) as a measure of chicken activity. Guo et al. [17] used machine vision to partition the chickens on the pen floor into functional areas such as drinking, feeding, and resting. By identifying the chickens and employing a BP neural network, they achieved spatial distribution recognition of the chickens, thereby approximating their activity. Existing automatic assessment methods for poultry activeness mostly focus on instantaneous images or short sequences of consecutive images. They typically use optical flow information and pixel occupancy as proxies for group activeness. However, these methods are susceptible to the influence of short-term external stimuli such as sampling actions or noise, and the static distribution of pixel points cannot track the chickens over an extended period, which diminishes the reliability of the measured chicken activity. On the other hand, there is limited research available in the field of meat duck farming on this topic.

The aim of this study was to utilize a video monitoring system for automatic segmentation and tracking of group-raised meat ducks in a captive environment. Deep learning techniques were used to process videos adaptively, obtaining the individual activeness of group-raised ducks. Subsequently, the impact of three different photoperiods on group-raised duck activeness was analyzed. Based on YOLOv8, the C2f-Faster-EMA module was introduced for network enhancement and to improve the performance of YOLOv8. Object recognition models, multi-object tracking (MOT) models, and instance segmentation models were constructed specifically for group-housed ducks. By identifying and tracking ducks in the videos, duck positions were obtained, and then through the segmentation task in the videos, the most representative keypoints of the ducks were extracted, and with the use of keypoints, duck movement distances were calculated and a visual representation of duck activeness was represented. In addition, rearing experiments were conducted based on different photoperiods to validate the effectiveness of this method. The outcomes of this research provide technical support for automatic analysis and assessment of duck behavior in commercial breeding facilities.

## 2. Materials and Methods

### 2.1. Experimental Design and Data Acquisition

The experiments were conducted in a meat duck farming facility located in the Liuhe District of Nanjing, China, from February to March 2023, spanning a total of 35 days. As shown in Figure 1b, the experiments took place in a closed environmental control warehouse, where two-tiered cages were set up, totaling 10 cages in all. An automated water supply system was installed to minimize human interference with the animals. Each cage contained 10 Cherry Valley meat ducks, totaling 100 ducks in mixed gender, with a stocking density of 12 ducks per square meter. Throughout the experiments, apart from the photoperiod, all other environmental factors and feed formulations were adjusted according to standard commercial meat duck farming practices.

The experiments randomly divided the ducks into three groups based on the photoperiod (24L:0D, 16L:8D, and 12L:12D), with each group having three cages for replication and one spare cage. Black opaque cloth was used to separate different photoperiod groups. Independent LED lights were installed in each cage, with the light intensity set at 20 lux. At the beginning of the experiments, the ducks were one day old and were given 5 days to adapt to the farming equipment and stabilize their physical condition. Throughout the entire experiments, there were three cases of abnormal duck mortality.

For video monitoring, IMX588 camera modules with USB output were used. One camera was centrally positioned above each cage, at a height of 70 cm from the cage floor, providing coverage of the entire cage. These cameras were connected to a PC via USB, and Python algorithms were employed to automatically capture images. To reduce data volume, cameras in 9 out of the 10 cages recorded continuous 6 min videos every hour, with a frame rate of 30 frames per second. Since the cameras lacked night vision capabilities, videos captured during periods of no illumination were considered invalid for data analysis. At the conclusion of the experiments, a total of 2658 valid video segments were collected, amounting to a total duration of 265.8 h.

### 2.2. Datasets Preparation

In this study, the OpenCV library was used to read videos and perform frame subsampling. Since the videos captured by the nine cameras contained information about the rearing of meat ducks under different lighting conditions during the 5- to 35-day-old age range, to ensure the uniformity of the dataset, 20 random videos were selected from each camera’s recordings. Furthermore, within each video, an average of 20 frames were evenly extracted. This resulted in a total of 3600 training images.

Before annotating the data, image enhancement techniques were applied. An FPGA algorithm [18] was used to adjust the image contrast, making the edge features of the meat ducks more distinct and prominent.

### 2.3. YOLOv8 Network Structure

The task of the model proposed in this study was to visually monitor the activeness of meat ducks using machine vision technology. This task requires high monitoring speed, especially when dealing with different moving groups of meat ducks, which poses challenges for segmentation and tracking accuracy. In commercial applications of object detection models, models based on YOLO (YOU ONLY LOOK ONCE) have advantages such as small model size, high training speed, and high accuracy [19]. They have been widely applied in agricultural farming scenarios. Considering the application context and accuracy requirements, this study is based on the latest iteration of the YOLO algorithm, the YOLOv8 model [20]. Specific optimizations were made to certain network structures to achieve relative light weightiness of the model and fast detection of complex meat duck targets, while ensuring accuracy. The meat ducks studied in this research undergo significant morphological changes during their growth period, requiring precise feature extraction. Therefore, YOLOv8x was selected as the primary recognition model.

The YOLOv8x network comprises four parts: input, backbone, neck module, and output. The input part includes mosaic data augmentation, adaptive anchor calculation, and adaptive grayscale padding. The backbone consists of Conv, C2f, and SPPF structures, with the C2f module being the primary module for learning residual features. This module is inspired by the efficient long-range attention network ELAN structure of YOLOv7 [21]. It enriches the gradient flow of the model through more branch cross-layer connections, forming a neural network module with stronger feature representation capabilities. The neck module adopts the PAN (path aggregation network) structure, enhancing the network’s ability to fuse features of objects at different scales. The output part decouples the classification and detection processes, including loss calculation and target detection box filtering. The loss calculation process includes positive and negative sample allocation strategies and loss calculation. YOLOv8x primarily uses the Task Aligned Assigner method, which selects positive samples based on the weighted results of classification and regression scores. The loss calculation includes two branches: classification and regression, with no objectness branch. The classification branch continues to use BCE loss, while the regression branch uses distribution focal loss and CIOU (complete intersection over union) loss functions.

The network structure of the improved YOLOv8x in this study is illustrated in Figure 2.

### 2.4. YOLOv8 Model Improvement Strategy

Constrained by the breeding strategy, the lighting conditions in duck houses are relatively dim, with light intensities of around 10–20 lux. In this study, a light intensity of 20 lux was set. Insufficient lighting leads to blurriness in images, making it challenging to discern individual ducks and boundaries within the duck group. Furthermore, as ducks continue to grow and enter the molting stage, the cage space becomes cramped and overcrowded, diminishing not only the cleanliness within the cages but also exacerbating the issue of ducks obstructing each other, resulting in cluttered backgrounds in images. These conditions introduce significant noise interference during image feature extraction by neural networks, thereby diminishing distant pixel dependencies.

Hence, the optimization strategy of this model primarily addressed two aspects: model computation light weighting and fine-grained feature extraction. On the one hand, it aimed to reduce the probability of missing and misclassifying small targets. On the other hand, it sought to address the challenge of target loss in complex environments.

#### 2.4.1. C2f-Faster Module

In YOLOv8, the backbone of the C3 module from YOLOv5 was replaced with the C2f module to achieve high-quality image feature extraction and downsampling. However, the C2f module also increased the model’s parameter count and complexity. To enhance the model’s training and inference speed, the bottleneck in the C2f module was replaced with the FasterNet [22] block, implementing efficient spatial feature extraction inspired by the concept of partial convolution (Pconv).

The C2f module, designed with inspiration from both the C3 module and the ELAN concept, utilizes multiple bottleneck operations to obtain richer gradient information. The bottleneck consists of two 1 × 1 convolution layers and one 3 × 3 convolution layer to perform dimension reduction, convolution, and dimension expansion on the input. However, this approach involves a significant number of floating point operations. To reduce the floating point operations per second (FLOPS), the FasterNet block introduces partial convolution (PConv), which simultaneously reduces memory access and computational redundancy. It applies a regular convolution to extract the spatial features from a portion of the input channels while leaving the rest unchanged. The FLOPS for PConv are calculated as follows:$$h\times w\times k^{2}\times c_{p}^{2}$$
where *h* and *w* represent the width and height of the feature map, respectively; *k* is the kernel size; and *c_p_* is the number of channels on which the convolution operates. Typically, *c_p_* is 1/4 of the number of channels used in a regular convolution. Therefore, the FLOPS of PConv are only 1/16 of those of a regular convolution.

Based on PConv, FasterNet was constructed, as shown in the upper right corner of Figure 2. FasterNet consists of four hierarchical stages, each of which is preceded by an embedding layer (a regular Conv 4 × 4 with stride 4) or a merging layer (a regular Conv 2 × 2 with stride 2) for spatial downsampling and channel number expansion. Each stage contains a stack of FasterNet blocks. Each FasterNet block comprises a PConv layer followed by two PWConv (or Conv 1 × 1) layers. Batch normalization and ReLU are employed as normalization and activation layers, effectively reducing the FLOPS.

#### 2.4.2. Efficient Multi-Scale Attention Module

To enhance pixel-wise dependencies, attention mechanism modules (CBAM [23], SE [24]) are introduced into convolutional neural networks, which have been proven to improve object detection and recognition. However, these attention modules involve a manual design with numerous pooling operations, significantly increasing computational requirements. For the meat duck breeding dataset, this model incorporated the efficient multi-scale attention module (EMA) [25] into the YOLOv8’s C2f-Faster module.

EMA is a novel and efficient multi-scale attention module that does not require dimension reduction. It achieves uniform distribution of spatial semantic features across each feature group by reshaping a portion of the channels into the batch dimension and grouping channel dimensions into multiple sub-features.

The EMA module primarily consists of three stages. The first is feature grouping, where for any given input feature map, EMA divides the feature map into multiple sub-features along the channel dimension to learn different semantics. Second, it conducts cross-spatial learning. Given the large local receptive fields of neurons that enable the collection of multi-scale spatial information, EMA uses three parallel routes to extract attention weight descriptors of the grouped feature maps. On the one hand, two encoded features in the image’s height direction are concatenated and share the same convolution without dimension reduction, resulting in two parallel routes obtaining different cross-channel interactive features. On the other hand, a separate branch captures local cross-channel interactions to expand the feature space. In this way, EMA encodes inter-channel information to adjust the importance of different channels while preserving precise spatial structure information into channels. Finally, EMA utilizes 2D global pooling operations to encode and output global spatial information for each branch’s output. The output results are multiplied using matrix dot-product operations, collecting different scale spatial information in the same processing stage. In this research, the EMA module was added to the forward propagation process of FasterNet, enabling the accurate extraction of spatial features for small objects.

Based on the original YOLOv8 network, this study replaced the C2f module in the backbone with the C2f-Faster-EMA module. This improvement was made to reduce the number of floating point operations during feature extraction in the backbone network. Additionally, an attention mechanism was incorporated into the forward propagation process, focusing the feature extraction process on enhancing the precision of small object detection. The inclusion of the attention mechanism increased the computational load of the model but was most effective during image feature extraction. Therefore, for the C2f module in the neck of YOLOv8, the EMA module was not added, and it was replaced with the C2f-Faster structure.

### 2.5. Method for Calculating Activeness of Meat Ducks

Activeness is one of the crucial indicators for assessing the efficiency of meat duck farming. Previous studies [15,16] have mostly relied on metrics such as distribution indices or pixel-wise mean displacement as indicators. However, these metrics lack reliable and intuitive foundations, and they are primarily focused on the collective behavior of meat duck populations, making it challenging to accurately assess individual duck activeness. To precisely calculate the activeness of meat ducks, this research proposed a calculation method that targeted the activity trajectories of group-raised meat ducks. Building upon the foundation of MOT of meat ducks, this method involved segmenting duck masks, extracting morphological centroids, and computing the activity trajectories and movement distances of individual ducks.

#### 2.5.1. MOT in Group-Raised Meat Ducks

The MOT model developed in this study for group-housed ducks consisted of two stages: object detection and tracking. The improved YOLOv8x model served as the target classification detector, enabling the extraction of individual ducks from images, and Bot-SORT [26] was employed as the tracking tool to monitor the IDs and positions of ducks at different time intervals.

Bot-SORT is a tracking-by-detection (TBD) paradigm-based object tracking method that utilizes the bounding boxes obtained from object detection for trajectory tracking. Bot-SORT enhances the robustness of detection results by integrating both motion and appearance information of detected objects. Currently, “Sort-Like” MOT methods draw inspiration from DeepSORT’s [27] aspect ratio estimation regression boxes, but this approach can decrease the precision of bounding box estimation. Bot-SORT modifies the seven-tuple state vector used in the tracking algorithm to describe object positions using width and height. It introduces a method for IoU and ReID-based cosine distance fusion, improving the match quality between detections and trajectories. Bot-SORT also addresses the possibility of camera movement by proposing a tracker based on camera motion compensation. However, in this research, where the camera position remained constant, tracking accuracy was further enhanced.

The MOT process for ducks in this study is as follows:(1)Calculate the confidence scores of meat duck detection boxes based on the improved YOLOv8x algorithm. Discard detection boxes with confidence scores below 0.1, categorize those with scores between 0.1 and 0.45 as low-confidence detection boxes, and classify those with scores above 0.45 as high-confidence detection boxes.(2)In the first matching step, match high-scoring detection boxes with previously tracked trajectories and obtain predicted target trajectories using Kalman filters.(3)In the second matching step, match low-scoring detection boxes with trajectories that were not matched in the first step.(4)Match the remaining detection boxes from both matching steps with newly appearing trajectories during the tracking process.(5)Retain the remaining tracking trajectories for 30 frames and attempt matching when the target box reappears; otherwise, discard the trajectory.

The MOT results for meat ducks is shown in Figure 3.

#### 2.5.2. Mask Segmentation and Centroid Coordinate Extraction for Meat Duck

After performing tracking, the results included meat duck detection box coordinates, IDs, confidence scores, and category information. The centroid coordinates calculated based on the detection boxes can represent the approximate relative positions of meat ducks. However, in practical farming scenarios, due to variations in meat duck postures, the centroid coordinates of detection boxes may be located at different parts of the meat duck or even outside the meat duck mask, which may lack representativeness. Therefore, in this study, on top of MOT, YOLOv8 was employed as the meat duck instance segmentation model to segment meat duck masks and compute centroid coordinates using morphological methods, making the obtained point coordinates more representative.

The concept of using YOLOv8 for instance segmentation combines object detection with instance segmentation to simultaneously detect and segment multiple object instances. Compared to traditional instance segmentation algorithms such as Mask RCNN [28] or U-Net [29], YOLOv8 instance segmentation offers faster speed and high accuracy. YOLOv8 utilizes a pre-trained semantic segmentation model to segment the candidate boxes obtained from object detection. It extracts features at different scales through a multi-scale feature pyramid network. Additionally, it employs an adaptive receptive field strategy by adjusting the convolution kernel size and stride to adapt to objects of different sizes, thereby improving instance segmentation performance.

In this study, meat duck contour centroids were calculated using morphological methods. For each meat duck mask, image binarization was performed based on the regression box obtained from object detection. Then, the mask contour was identified, and the image moment ‘M’ was calculated to obtain the centroid coordinates:$$Cx=\frac{M10}{M00},\;Cy=\frac{M01}{M00}$$

The obtained centroid coordinates were relative coordinates with respect to the regression box. For group-raised meat duck images, the actual positions of centroid coordinates for each individual meat duck are shown as follows:$$\begin{array}{c} Sx=Zx+Cx\\ Sy=Zy+Cy \end{array}$$
where *S_x_* and *S_y_* respectively represent the *x*- and *y*-axis coordinates of the meat duck relative to the image, respectively; while *Z_x_* and *Z_y_* denote the left-bottom *x*- and *y*-axis coordinates of the meat duck mask’s regression box, respectively.

The mask segmentation and centroid coordinate results for meat ducks are shown in Figure 4.

### 2.6. Algorithm Platform

The hardware utilized for model training in this study consisted of a 12th Gen Intel® Core™ i9-12900K processor operating at a clock frequency of 3.2 GHz, with 128 GB of RAM and an NVIDIA GeForce RTX 4090 GPU. The operating system employed was Windows 10. Python served as the development language, and development and training were conducted using the Ultralytics framework. The model underwent 100 training epochs with an initial learning rate of 0.01. After multiple rounds of hyperparameter tuning, the model with the highest recognition accuracy was selected as the final model.

## 3. Results and Discussion

### 3.1. Performance Evaluation of Machine Vision Algorithm Models

Since this paper presents an improved model built upon the YOLOv8 framework, its functionality can be divided into three main parts: object detection, MOT, and instance segmentation. Due to the primary emphasis of this study on evaluating the performance of object detection and MOT networks, the performance assessment was directed toward these aspects. As for the instance segmentation task, the YOLOv8 framework was utilized solely for obtaining more precise centroid coordinates. Consequently, the performance of instance segmentation will not be discussed in this paper.

For the object detection phase, the paper employs the mean average precision (mAP), recall (R), and model parameter count as evaluation metrics.

For the accuracy of MOT, this paper utilizes the multiple object tracking accuracy (*R_MOTA_*) as the evaluation metric, calculated using the following formula:$$R_{MOTA}=1-\frac{\sum_{t}\left(m_{t}+f_{t}+s_{t}\right)}{\sum_{t}g_{t}}$$
where *m_t_* represents the number of missed detections of meat ducks in the video, *f_t_* represents the number of false detections of meat ducks in the video, *s_t_* represents the number of ID switches, *g_t_* represents the actual number of targets in the video sequence, and *t* represents the number of test videos.

### 3.2. Experimental Comparison of Model Performance

#### 3.2.1. Object Detection Performance for Improved YOLOv8

To visually demonstrate the impact of the enhancement module on model performance, YOLOv5 was used for comparison with our model. In YOLOv5, a Faster-EMA module structurally identical to the one presented in this study was inserted into the C3 module. With the same dataset and hyperparameters, the Yolov5 network improved with the C3-Faster-EMA module, the unimproved YOLOv8, and the YOLOv8 network improved with the C2f-Faster-EMA module were trained. The training performance is shown in below.

As illustrated in Figure 5 and Table 1, when considering the overall performance, the improved YOLOv8x outperformed the other two networks across most metrics. In terms of training box loss, the improved YOLOv8x achieved the lowest loss value, registering at 0.275, followed by the improved YOLOv5x and the original YOLOv8x with loss values of 0.302 and 0.340, respectively. Throughout the training epoch, due to structural similarities, the training losses exhibited clear trends, indicative of stable training. Upon the incorporation of the Faster-EMA module, YOLOv5’s performance exceeded that of YOLOv8, implying that the enhancement module effectively enhanced the network’s precision in segmenting smaller objects. Nevertheless, YOLOv8, which initially demonstrated superior performance, maintained its lead even with the enhancement module.

The values of mAP and Recall further supported the above conclusions. The mAP@50-90 for the improved YOLOv8 was 0.979, which was 1.45% and 2.73% higher than that of the improved YOLOv5 and YOLOv8, respectively. This demonstrated that the improved YOLOv8 performed better in duck recognition and localization, although the improvement was not substantial. This may be because the duck images had clear features during preprocessing, and the breeding environment was relatively uniform. To adapt to the actual breeding environment, this study selected the YOLO model with the highest number of layers and largest size (YOLOv8x), which prevented significant performance gaps between the networks. Therefore, all models achieved high mAP values. This was also evident in the Recall results, where all models had Recall values above 0.99, indicating good generalization performance in target acquisition.

In terms of frames per second (FPS), the improved YOLOv5 exhibited the fastest inference speed, with an average inference time of 56.9 ms. On the other hand, the improved YOLOv8 had the slowest inference speed, with an average inference time of 67 ms, which was 15.0% slower than the improved YOLOv5. The original YOLOv8 fell between the two in terms of inference speed. These results indicated that the insertion of the Faster-EMA module increased the computational load of the network, leading to reduced inference speed. YOLOv5, being an earlier release with a mature algorithm structure, outpaced YOLOv8 in speed. However, since this study required a further MOT task on top of object detection, demanding higher network accuracy, the slight increase in inference time for the improved YOLOv8 (only 0.01 s slower than the improved YOLOv5) had minimal practical impact. Therefore, taking all factors into account, using the improved YOLOv8 network for duck recognition in this study was a more reasonable choice.

#### 3.2.2. MOT Performance for Improved YOLOv8

The improved YOLOv8 model was employed as the detector for tracking ducks in the videos. Three different tracking models, ByteTrack [30], Bot-SORT, and DeepSORT, were used to track the ducks in the videos. The tracking performance of these three tracking models is presented in Table 2:

The results presented in Table 2 indicate that when using the improved YOLOv8 as the detector, Bot-SORT, DeepSORT, and ByteTrack all exhibited high tracking accuracy. This wa primarily attributed to the detector’s high detection precision, and the fact that the video scene remained consistently stable with all ducks continuously within view. Cases of ducks leaving the frame and returning were rare, enabling each tracking module to fully leverage its performance potential. However, the Bot-SORT module demonstrated superior performance among the three, achieving a MOTA of 85.1%. This was 8.2% higher than DeepSORT and 4.5% higher than ByteTrack. Cases of tracking errors typically arose when ducks were densely clustered, making it challenging to distinguish individual ducks, resulting in overlapping detection boxes and extended periods of tracking information loss. This underscored the need for further improvements in detection accuracy. Additionally, the three trackers all had increased execution times, while Bot-SORT had the highest per-frame computational cost and ByteTrack the lowest. Compared to the detection speed of the detector, the per-frame execution time almost doubled after adding the trackers. However, due to MOTA being more crucial when assessing the model’s value, this research still employed Bot-SORT as the model’s tracker.

Figure 6 illustrates examples of tracking errors across consecutive frames. These instances of tracking errors emphasize the importance of enhancing the precision of the detector to better handle scenarios where ducks are closely grouped, further improving tracking accuracy.

### 3.3. Impact of Photoperiod on the Activeness of Meat Ducks

In this study, by tracking the centroid coordinates of group-raised meat ducks, we obtained the relative positions of each meat duck at each moment, and centroid coordinates were aligned to obtain the movement trajectories of meat ducks. Thus, the activeness of meat ducks during a certain time period could be defined as the length of their movement. Then, as described in Section 2.1, this study conducted 35-day-long breeding experiments, dividing the meat ducks into three photoperiod groups with each group having three sets of repeated experiments and each group consisting of 10 meat ducks.

This study employed the single-frame average displacement (SFAD) of meat ducks as an indicator to assess the impact of the photoperiod on meat duck activeness. The SFAD was calculated for meat ducks of different age groups under various photoperiods. The calculation formula of meat duck SFAD is shown as follows:
$$L=\sum_{j=1}^{k}\frac{\sum_{i=2}^{n}\sqrt{{\left(x_{ki}-x_{k(i-1)}\right)}^{2}+{\left(y_{ki}-y_{k(i-1)}\right)}^{2}}}{n}$$
where *L* represents the SFAD of meat ducks in the video, *k* is the meat duck index, *x_i_* and *yi* represent the centroid coordinates of meat duck *k* in the *i*-th frame, and *n* is the number of frames in which meat duck *k* can be tracked in the video.

Due to ID switching in the MOT process in the video, the same tracked meat duck could have different IDs. To ensure the accuracy of the calculation results in this study, when an ID was switched, only the coordinate distances over consecutive frames would be computed for a meat duck under the original ID. For the new ID resulting from the switch, if the number of frames associated with the new ID was less than 60 frames, equivalent to two seconds, it was considered invalid and removed. Since the SFAD was weighted based on the number of frames to calculate the total movement of meat ducks, ID switching had minimal impact on the calculation results.

The SFAD data of meat ducks in nine cages were calculated. The results are shown in Figure 7, where each data point represents the SFAD for all meat ducks in the corresponding enclosure over a 6 min recording video.

Figure 7 indicates that from February 28th (6 days old) to March 25th (31 days old), the SFAD of meat ducks in each cage gradually increased with age, regardless of the photoperiod. Under different photoperiods, the meat duck SFAD in each cage gradually increased with age, and the growth rate also increased. This indicated that, in comparison to the photoperiod, age had a greater impact on the activeness of meat ducks. On March 15th (after 20 days of age), there was a noticeable dispersion in the points of SFAD, and the range gradually expanded. This indicated that the activeness of meat ducks in the later stages of growth was significantly higher than that in the earlier stages. The actual video observations showed a significant decrease in the clustering behavior of meat ducks after 20 days of age, leading to an increase in activeness. This might be attributed to the significant increase in heat production by meat ducks as they grow, reducing the need for huddling for warmth and thus increasing their activeness.

To further analyze the impact of the photoperiod on meat duck activeness, the SFAD of meat ducks in nine cages across the three photoperiod groups was statistically summarized, as shown in Table 3.

As shown in Table 3, the meat duck group with the 24L:0D photoperiod exhibited the highest meat duck activeness, with an SFAD value of 1.003. The group with the 12L:12D photoperiod had the second highest activeness, with an SFAD value of 0.983. In contrast, the meat duck group with the 16L:8D photoperiod showed the lowest activeness at only 0.906, representing a decrease in activeness of 9.63% compared to the 12L:12D photoperiod group. Regarding the relationship between photoperiod and SFAD, it was evident that the duration of lighting and meat duck activeness did not follow a linear pattern. Ducks with the 24L:0D photoperiod exhibited the highest activeness, aligning with the current practice of a 24L:0D photoperiod in commercial meat duck farming in China. However, with a 12L:12D photoperiod from 8 AM to 8 PM, which corresponds closely to the natural photoperiod, meat duck activeness was only slightly reduced by 1.99% compared to the 24L:0D photoperiod group.

The experimental results indicated that the meat duck group with the 12L:12D photoperiod, which is similar to natural lighting, showed comparable activeness to the group with the 24L:0D photoperiod used in commercial farming. In contrast, the group provided with the 16L:8D photoperiod exhibited significantly lower activeness than the first two groups. This difference may be attributed to the fact that the 12 h lighting duration aligns better with the growth characteristics of meat ducks, allowing them to exhibit optimal physiological behavior. When the lighting duration was increased to 16 h, the physiological functions of the meat ducks may have decreased, resulting in a reduction in SFAD. However, with a further increase in lighting duration to 24 h, the meat ducks remained under constant light stress, which led to increased activeness.

Taken together, a 24L:0D photoperiod may enhance meat duck stress and maintain a high activeness level in the duck population, but it can also potentially affect their physiological state. In contrast, the activeness of meat ducks with the 12L:12D photoperiod was not significantly different from those with the 24L:0D photoperiod, but this photoperiod could reduce energy consumption by half and double the lifespan of lighting equipment.

To investigate the effects of the photoperiod on meat ducks at different age stages, the study divided the experimental period into four stages, representing the age ranges of 6–13 days, 14–19 days, 20–25 days, and 26–31 days, respectively. Figure 8 illustrates the relationship between photoperiod and meat duck activeness, where each data point represents the mean value of the SFAD for the corresponding camera during that age stage.

As shown in Figure 8, it is observed that the activeness of meat ducks did not show a significant increase during the 6–19 day age stage. In fact, there were instances of decreased activeness in several groups of meat ducks during this period. However, starting from the 19th day, the activeness of all groups of meat ducks exhibited a significant increase, albeit at different growth rates.

Before the 19th day, compared with the other two groups, meat ducks with the 12L:12D photoperiod (Camera6, Camera7, and Camera8) had lower activeness. However, by the end of the experiments, when the ducks were 31 days old, their average activeness surpassed that of the other photoperiod groups. Meat ducks with the 16L:8D photoperiod (Camera3, Camera4, and Camera5) had activeness during the 6–19 day age range that fell in between that of the other two groups. However, by the end of the experiments, their average activeness was the lowest among the three groups. Meat ducks with the 24L:0D photoperiod (Camera1, Camera2, and Camera3) initially exhibited the highest activeness, but their growth rate of activeness slowed down in the later stages of the experiments.

Taking into account the experimental results from Figure 7 and Table 2, it can be observed that over the entire experimental period, the SFAD in the 12L:12D photoperiod duck group was slightly lower than that in the 24L:0D photoperiod group. Additionally, the activeness of meat ducks in the 12L:12D photoperiod group showed a pattern of initially lower activeness during the early growth phase, followed by higher activeness in the later stages. The average growth rates of activeness over the three time periods were 10.4%, 15.5%, and 17.7% higher than those in the 24L:0D photoperiod group, respectively. It can be inferred that as the experiments continued, the activeness of meat ducks in the 12L:12D photoperiod group would likely surpass that of meat ducks in the 24L:0D photoperiod group. This provides further evidence that the high activeness observed in the 24L:0D photoperiod group was the result of prolonged light exposure-induced stress. In the early stages of meat duck growth, their activeness was significantly higher than that in the other photoperiod groups. However, as the meat ducks grew, the continuous light exposure led to a decrease in their physiological abilities. In contrast, meat ducks receiving natural lighting had better welfare conditions, which is why they displayed higher activeness in the later stages of growth. In addition, meat ducks subjected to the 16L:8D photoperiod consistently exhibited lower activeness and slower activity growth throughout the entire growth period.

## 4. Discussion

In order to explore the impact of the photoperiod on meat duck activeness, this study constructed a machine vision-based calculation model for calculating the activeness of meat ducks. By continuously tracking the position of meat ducks, the movement distance of each duck was recorded to calculate the activeness. This method was not only more intuitive than existing activeness assessment methods but also not limited by the number of meat ducks, providing a feasible technical means for assessing the activeness and welfare of meat ducks in group breeding.(1)Based on YOLOv8, the C2f-Faster-EMA module was introduced to optimize the backbone network, neck, and head. The training performance of the improved YOLOv8, YOLOv8, and the improved YOLOv5 on the same meat duck dataset was compared. The results showed that the improved YOLOv8 in this paper had better performance in the complex environment of group-raised meat ducks. The mAP50-90 performance of the target detection was 0.979, which was 1.45% and 2.73% higher than that of YOLOv8 and the improved YOLOv5, respectively.(2)Based on the improved YOLOv8 model, a MOT model based on Bot-SORT was constructed to achieve real-time tracking of group-raised meat ducks. With the same detector, the tracking performance of two other MOT model algorithms, DeepSORT and ByteTrack, was compared. The results showed that the improved YOLOv8+Bot-SORT model constructed in this paper had better tracking performance with a MOTA of 85.1%, which was 8.2% and 4.5% higher than that of DeepSORT and ByteTrack, respectively.(3)This study built an instance segmentation model based on the improved YOLOv8, and used OpenCV technology to extract the centroid coordinates of the duck masks instead of the center point of the multi-target tracking regression box as the representative feature point of the meat duck, which achieved higher tracking accuracy of the meat duck’s position.(4)Based on the constructed model for calculating the activeness of meat ducks, this study used the SFAD of meat ducks as the indicator for calculating activeness. A monitoring experimental platform for the activeness of meat ducks was built, and a 35-day breeding experiment was conducted to verify the effects of three photoperiods (24L:0D, 16L:8D, and 12L:12D) on the activeness of meat ducks. The results showed that the model constructed in this study could effectively calculate the activeness of meat ducks. During the experimental period, the meat duck group with the 24L:0D photoperiod had the highest SFAD, which could be considered to have the highest activeness. However, compared with the meat duck group with the 12L:12D photoperiod, it was found that the 24L:0D photoperiod was likely to be a greater stimulus on meat ducks, resulting in a significant decrease in the growth rate of meat duck activeness in the later stage of breeding. The meat duck group with a natural lighting duration (12L:12D photoperiod) showed lower activeness in the initial stage of breeding, but with increasing age, their activeness increased the most, and the overall activeness was close to that of the 24L:0D photoperiod meat duck group, with less stress. Therefore, in commercial farming, from the perspective of meat duck activeness, a 12L:12D photoperiod is more beneficial for meat duck growth than a 24L:0D photoperiod.

## 5. Conclusions

The main objective of this study was to construct a more intuitive model for assessing the activeness of meat ducks, with the goal of assisting in the development of a welfare monitoring system for poultry farming. By integrating machine vision models such as object detection, MOT, and instance segmentation, we developed a non-contact assessment method of meat duck activeness in this research. The results indicated that the improved YOLOv8-C2f-Faster-EMA model achieved an object detection performance of 0.979, with a MOTA of 85.1% when using the Bot-SORT tracker. This model was capable of tracking the trajectories and calculating the activity of group-reared meat ducks in caged conditions. Additionally, we conducted breeding experiments to investigate the impact of the photoperiod on meat duck activeness. The results of the rearing experiments demonstrated that, when using meat duck activity as the criterion, meat ducks under a photoperiod of 24L:0D exhibited the highest activity. However, meat ducks under a photoperiod of 12L:12D displayed greater vitality and growth potential. Our study provides technical and theoretical support for optimizing animal breeding on farms.

While the activeness computational method and breeding experiments developed in this study achieved the intended goals, there are still some issues to address. Firstly, due to the absence of thermal infrared cameras, the activeness data of the two groups of meat ducks with photoperiods of 16L:8D and 12L:12D could not be collected under dark conditions. Therefore, the data presented in this paper can only partially explain the influence of the photoperiod on meat duck activeness. Secondly, this study only focused on the duration of the photoperiod and its impact on meat duck activeness, without discussing the relationship between light photoperiod and animal production indicators such as feed-to-meat ratio and defect rate. Consequently, the conclusions regarding the photoperiod and meat duck activeness from this study cannot be straightforwardly applied to production settings. Lastly, the image capture height of the experimental platform for the activeness monitoring model constructed in this study was 20 cm higher than that of commercial poultry cages. As a result, it cannot be directly used in commercial cages. Nevertheless, through this experiment, we discovered that even with a lower photo capture height, effective images of meat ducks could still be captured. Therefore, by further improving the model and experimental setup, our equipment and technology will contribute to enhancing the level of automation in Chinese meat duck farming in the future.

## Figures and Tables

**Figure 1 animals-13-03520-f001:**
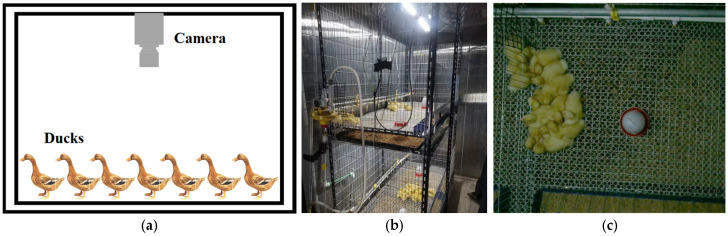
Data acquisition methods: (**a**) Schematic diagram of camera layout; (**b**) On-site representation of farming facilities; (**c**) Examples of collected images.

**Figure 2 animals-13-03520-f002:**
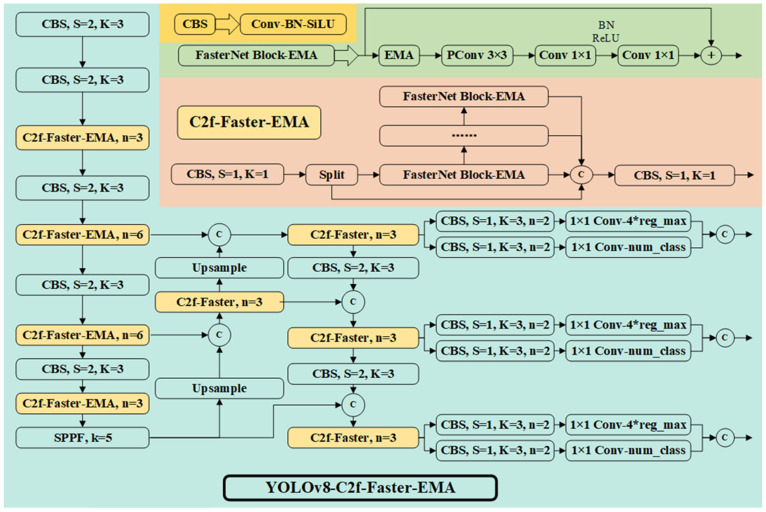
YOLOv8 network architecture enhanced with C2f-Faster-EMA.

**Figure 3 animals-13-03520-f003:**
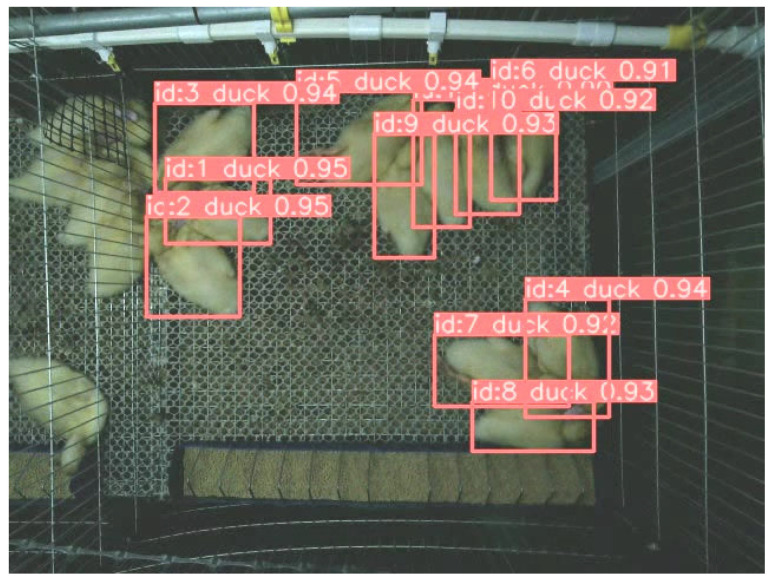
The MOT results for meat ducks.

**Figure 4 animals-13-03520-f004:**
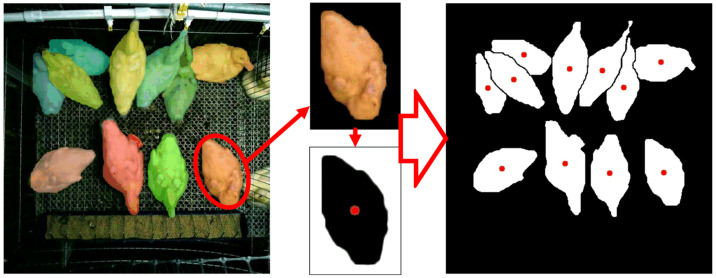
The mask segmentation and centroid coordinate results for meat ducks.

**Figure 5 animals-13-03520-f005:**
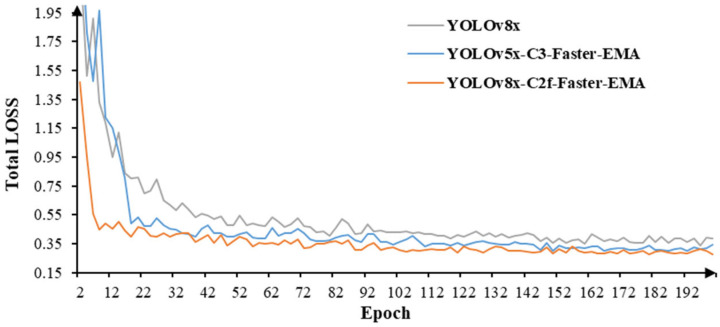
The training losses of three models.

**Figure 6 animals-13-03520-f006:**
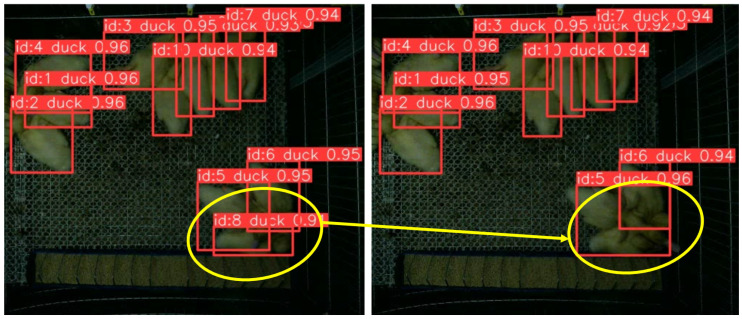
Tracking errors across consecutive frames.

**Figure 7 animals-13-03520-f007:**
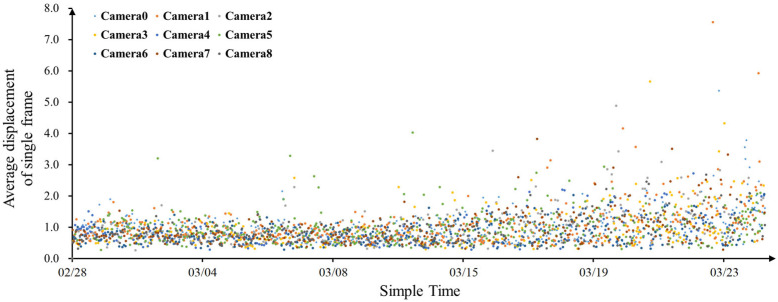
The SFAD of meat ducks in nine cages.

**Figure 8 animals-13-03520-f008:**
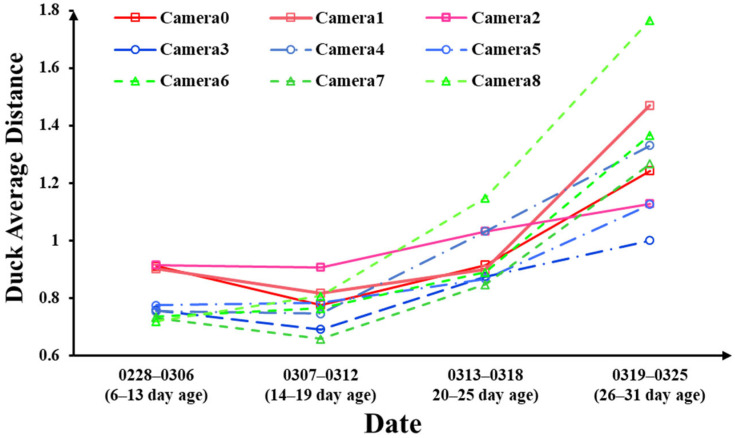
The SFAD of meat ducks in different age stages.

**Table 1 animals-13-03520-t001:** The object detection performance of three models.

Model	mAP@50-95	Recall	Time (ms/f)
YOLOv8x	0.953	0.996	66.4
YOLOv5x + C3-Faster-EMA	0.965	0.997	56.9
YOLOv8x + C2f-Faster-EMA	0.979	0.997	67.0

**Table 2 animals-13-03520-t002:** The MOT performance of three models.

Tracking Model	YOLOv8 + DeepSORT	YOLOv8 + ByteTrack	YOLOv8 + Bot-SORT
*R_MOTA_*	0.769	0.806	0.851
Execution time (ms/f)	103.5	88.4	121.8

**Table 3 animals-13-03520-t003:** The SFAD of meat ducks under different photoperiods.

Photoperiod	Camera ID	SFAD (Caged)	SFAD (Photoperiod)
24L:0D	Camera0	0.975	1.003
	Camera1	1.041	
	Camera2	0.995	
16L:8D	Camera3	0.856	0.906
	Camera4	0.953	
	Camera5	0.909	
12L:12D	Camera6	1.002	0.983
	Camera7	0.863	
	Camera8	1.085	

## Data Availability

Data are contained within the article.

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
