# Peer review of "Regulation of Meat Duck Activeness through Photoperiod Based on Deep Learning"

_animals, 2023, doi:10.3390/ani13223520_

Round 1

Reviewer 1 Report

Comments and Suggestions for Authors

Dear Author(s),

The manuscript should revised according to the comment.

Best Regards.

REVIEWER REPORT

Abstract section

The abstract section should begin with the importance of this study. In addition, at the end of the abstract section should suggest something for the researchers/readers.

Introduction section

Line 52: There is a need for reference.

Line 91-104: A lot of "we" expressions are used in almost every sentence, even within the sentence, and I think this situation disrupts the fluency of the sentence. It can be conveyed more effectively. I suggest it be rewritten.

Material and Methods section

Line 114: In this process, could you give a piece of information about the content of the ducks' rations?

Line 116: Which experimental design did you adopt for this study? Please give information about this aspect.

Line 126: Please provide information about your computer's specifications and citation for “Python”.

Results and Discussion section

Line 340-347: The paragraph should be mentioned in the Material and Methods section. In this context, the beginning of the results and discussion section needs another introduction sentence.

Line 375 and Line 432:  There are two “Figure 5”. Please check the order of Figures and Tables. Also, please correct the confusion in the text caused by this confusion.

Line 546-588: Please discuss the results of the current study with other studies. The discussion section should be rewritten. The available information in the Discussion section is also critical. I think this information should be transferred to the end of the results or conclusion sections.

Conclusion section

Line 589-612: The conclusion should be reconstructed, considering the statements in my previous comment.

Author Response

REVIEWER REPORT

Abstract section

  1. The abstract section should begin with the importance of this study. In addition, at the end of the abstract section should suggest something for the researchers/readers.

Answer: Authors are thankful for this comment. The abstract has been enhanced by introducing the importance of this study at the beginning and by providing recommendations at the end.

Introduction section

  1. Line 52: There is a need for reference.

Answer: Thanks for the suggestion. The corresponding reference (Reference 6) has been added to the paper.

  1. Line 91-104: A lot of "we" expressions are used in almost every sentence, even within the sentence, and I think this situation disrupts the fluency of the sentence. It can be conveyed more effectively. I suggest it be rewritten.

Answer: Thanks for the suggestion. The language of the paper has been optimized.

Material and Methods section

  1. Line 114: In this process, could you give a piece of information about the content of the ducks' rations?.

Answer: The authors appreciate the comments from the reviewer. The authors believe that the information regarding rations has limited relevance to the current study, and including excessive unrelated information may affect the paper's structure. Therefore, it was not included in the paper. However, the authors are willing to inform the reviewer that the ducks were fed with regular commercial meat duck feed produced by a duck feed manufacturer in Nanjing, China. Different types of feed were used for ducks at various growth stages, including chick feed, intermediate feed, and grower feed. Thanks again.

  1. Line 116: Which experimental design did you adopt for this study? Please give information about this aspect.

Answer: Thanks for the comment. The information has been given.

  1. Line 126: Please provide information about your computer's specifications and citation for “Python”.

Answer: The authors appreciate the comments. The computer information were shown in the beginning of Sec.2.6

Results and Discussion section

  1. Line 340-347: The paragraph should be mentioned in the Material and Methods section. In this context, the beginning of the results and discussion section needs another introduction sentence.

Answer: Thanks for the comments. The paragraph's placement has been modified.

  1. Line 375 and Line 432: There are two “Figure 5”. Please check the order of Figures and Tables. Also, please correct the confusion in the text caused by this confusion.

Answer: Thanks for the comments. All errors have been corrected.

  1. Line 546-588: Please discuss the results of the current study with other studies. The discussion section should be rewritten. The available information in the Discussion section is also critical. I think this information should be transferred to the end of the results or conclusion sections.

Answer: Thanks for the comments. The authors have compared several similar models and enhancement algorithms in the discussion and have proposed the optimal model. Conclusive statements from the discussion section have been partially moved to the conclusion. Once again, we appreciate the reviewer.

Conclusion section

  1. Line 589-612: The conclusion should be reconstructed, considering the statements in my previous comment.

Answer: Thanks for the comments. The conclusion should be reconstructed.

Reviewer 2 Report

Comments and Suggestions for Authors

line 39, 117, correct the "12D:12D" to "12L:12D".

line 422 check the number 9.6% it might be 8.2%.

line 423 check the number 5.2% it might be 4.5%.

line 427 Figure 9 might be Figured 5

Figure 1. (a) The Chinese character might be translated to English 

Author Response

Comments and Suggestions for Authors

  1. line 39, 117, correct the "12D:12D" to "12L:12D".

Answer: Authors are thankful for this comment. All errors have been corrected.

  1. line 422 check the number 9.6% it might be 8.2%.

Answer: Authors are thankful for this comment. The error has been corrected.

  1. line 423 check the number 5.2% it might be 4.5%.

Answer: Authors are thankful for this comment. The error has been corrected.

  1. line 427 Figure 9 might be Figured 5.

Answer: Authors are thankful for this comment. The error has been corrected.

  1. Figure 1. (a) The Chinese character might be translated to English.

Answer: Authors are thankful for this comment. The image has been corrected

Reviewer 3 Report

Comments and Suggestions for Authors

1.) In Figure 1. (a), please add the English term of the camera in the Fig.

2.) In Figure 7, there are several problems for authors to express their idea. Please modify the figure and the paragraph.

2.1 There is no unit for Duck Average Distance. Please add the unit for readers to evaluate the activeness of ducks.

2.2 And the date axis could not be easy to connect the date with the day age. Please add the day age in the figure to compare with the date. 

2.3 There are three photoperiod sets by using different sets of cameras. Please add the notification in the Figure and replace the symbols and the line formats of the cameras. They are all in  It will be really difficult to recognize the camera number by the black printing.

One interesting point, maybe we can have more discussion on it. The 12L:12D photoperiod was from 8 AM to 8 PM. Why authors did not set the time from 5 AM to 5 PM? It is probably more similar as the natural environment. I am thinking about the sun light affection.

Others are in good shape of the article.

Author Response

Comments and Suggestions for Authors

  1. In Figure 1. (a), please add the English term of the camera in the Fig.

Answer: Authors are thankful for this comment. The error has been corrected.

  1. In Figure 7, there are several problems for authors to express their idea. Please modify the figure and the paragraph.

2.1 There is no unit for Duck Average Distance. Please add the unit for readers to evaluate the activeness of ducks.

Answer: Thanks for the suggestion. Since the distances described in the image represent pixel displacement rather than actual physical distance units, there are no specific physical distance units. Thank you once again for the suggestion.

2.2 And the date axis could not be easy to connect the date with the day age. Please add the day age in the figure to compare with the date.

Answer: Authors are thankful for this comment. The error has been corrected.

2.3 There are three photoperiod sets by using different sets of cameras. Please add the notification in the Figure and replace the symbols and the line formats of the cameras. They are all in. It will be really difficult to recognize the camera number by the black printing.

Answer: Authors really appreciate this comment. The errors have been corrected.

One interesting point, maybe we can have more discussion on it. The 12L:12D photoperiod was from 8 AM to 8 PM. Why authors did not set the time from 5 AM to 5 PM? It is probably more similar as the natural environment. I am thinking about the sun light affection.

Answer: Thank you for the reviewer's feedback. Indeed, the timing of light exposure is an important consideration, but introducing this variable would significantly increase the workload and the number of experimental animals. On the other hand, since the ducks are raised in a fully enclosed environment, external light does not have any impact on them. Therefore, the rearing environment can be considered to be in a time zone where sunrise occurs three hours later than the local time. In future research, we will further investigate the impact of the light starting time. Once again, we appreciate the reviewer's feedback

Reviewer 4 Report

Comments and Suggestions for Authors

A duck activeness estimation method based on machine vision technology involves tracking the movement of group-raised ducks over a 6-minute period. The average displacement of all ducks within each frame is used as an indicator to measure activeness, resulting in more accurate results. Based on improved YOLOv8x, BoT-SORT and instance segmentation, the author designed a non-contact assessment of meat duck activeness. However, there are several issues which need to be fixed properly. Consequently, I would like to recommend this work publication after major revision.

(1)  There were too few references in Introsuction, and few descriptions of the problems with the methods proposed by previous researchers.

(2) The second sentence of the third paragraph in Introsuction could be added to relevant references.

(3) The improvement of the model towards lightweight is mentioned later, so the advantages of the model proposed in this paper compared with others can be mentioned in the introduction.

(4) In abstrast, Section 2.1, "(24L: 0D, 16L: 8D, 12D: 12D,)", there seems to be an error in "12D", and please state that L and D represent daylight time and darkness time, respectively.

(5) Some agriculture-related references can be added to the fourth sentence of the first paragraph of chapter 2.3, and the references of YOLOv7 can be added to the fourth sentence of the second paragraph, and the formula of YOLOv8 loss function is listed. The third paragraph says that Figure 2 is the structure of YOLOV8x, which should be the improved structure, please explain.

(6) Please add references to YOLOv5 in paragraph 1 of Section 2.4.1.

(7) Please center the formula in the text.

(8) Please add references after CBAM and SE in the first sentence of paragraph 1 of Section 2.4.2.

(9) Please add references after DeepSort in the third sentence of paragraph 2 of Section 2.5.1.

(10) Please add references after Mask RCNN and U-Net in the second paragraph of Section 2.5.2.

(11) In Figure 5 and Table 1, write the model in the same order as the columns.

(12) When drawing picture 7, the line color of the picture is not enough, which can be described with the same color + different shape patterns.

(13) The improved YOLOv8x mentioned many times in the article can be replaced with YOLOv8x+C2f-Faster-EMA.

(14) The advantages and contributions of the model are less described in the conclusion of the paper.

(15) Please change DeepSort and BotSort in the article to DeepSORT and Bot-SORT respectively.

Comments on the Quality of English Language

A duck activeness estimation method based on machine vision technology involves tracking the movement of group-raised ducks over a 6-minute period. The average displacement of all ducks within each frame is used as an indicator to measure activeness, resulting in more accurate results. Based on improved YOLOv8x, BoT-SORT and instance segmentation, the author designed a non-contact assessment of meat duck activeness. However, there are several issues which need to be fixed properly. Consequently, I would like to recommend this work publication after major revision.

(1)  There were too few references in Introsuction, and few descriptions of the problems with the methods proposed by previous researchers.

(2) The second sentence of the third paragraph in Introsuction could be added to relevant references.

(3) The improvement of the model towards lightweight is mentioned later, so the advantages of the model proposed in this paper compared with others can be mentioned in the introduction.

(4) In abstrast, Section 2.1, "(24L: 0D, 16L: 8D, 12D: 12D,)", there seems to be an error in "12D", and please state that L and D represent daylight time and darkness time, respectively.

(5) Some agriculture-related references can be added to the fourth sentence of the first paragraph of chapter 2.3, and the references of YOLOv7 can be added to the fourth sentence of the second paragraph, and the formula of YOLOv8 loss function is listed. The third paragraph says that Figure 2 is the structure of YOLOV8x, which should be the improved structure, please explain.

(6) Please add references to YOLOv5 in paragraph 1 of Section 2.4.1.

(7) Please center the formula in the text.

(8) Please add references after CBAM and SE in the first sentence of paragraph 1 of Section 2.4.2.

(9) Please add references after DeepSort in the third sentence of paragraph 2 of Section 2.5.1.

(10) Please add references after Mask RCNN and U-Net in the second paragraph of Section 2.5.2.

(11) In Figure 5 and Table 1, write the model in the same order as the columns.

(12) When drawing picture 7, the line color of the picture is not enough, which can be described with the same color + different shape patterns.

(13) The improved YOLOv8x mentioned many times in the article can be replaced with YOLOv8x+C2f-Faster-EMA.

(14) The advantages and contributions of the model are less described in the conclusion of the paper.

(15) Please change DeepSort and BotSort in the article to DeepSORT and Bot-SORT respectively.

Author Response

A duck activeness estimation method based on machine vision technology involves tracking the movement of group-raised ducks over a 6-minute period. The average displacement of all ducks within each frame is used as an indicator to measure activeness, resulting in more accurate results. Based on improved YOLOv8x, BoT-SORT and instance segmentation, the author designed a non-contact assessment of meat duck activeness. However, there are several issues which need to be fixed properly. Consequently, I would like to recommend this work publication after major revision.

  1. There were too few references in Introsuction, and few descriptions of the problems with the methods proposed by previous researchers.

Answer: Authors are thankful for this comment. The section has been supplemented in the paper according to the recommendations

  1. The second sentence of the third paragraph in Introsuction could be added to relevant references.

Answer: Thanks for the suggestion. The conclusion in this sentence is self-evident, and thus, no relevant papers were found for citation. Thank you once again for the feedback.

  1. The improvement of the model towards lightweight is mentioned later, so the advantages of the model proposed in this paper compared with others can be mentioned in the introduction.

Answer: Thanks for the suggestion. Due to the results show that the model's inference time per image increased with the addition of C2f-Faster-EMA, it did not achieve model lightweighting. Therefore, after discussions among the authors, it has been decided not to make any modifications in this regard.

  1. In abstrast, Section 2.1, "(24L: 0D, 16L: 8D, 12D: 12D,)", there seems to be an error in "12D", and please state that L and D represent daylight time and darkness time, respectively.

Answer: Authors are thankful for this comment. The error has been corrected.

  1. Some agriculture-related references can be added to the fourth sentence of the first paragraph of chapter 2.3, and the references of YOLOv7 can be added to the fourth sentence of the second paragraph, and the formula of YOLOv8 loss function is listed. The third paragraph says that Figure 2 is the structure of YOLOV8x, which should be the improved structure, please explain.

Answer: Authors really appreciate this comment. The errors have been corrected. In addition, the loss function of YOLOv8 has already been explained in the text, and the keypoint of this paper is on the improvements made to YOLOv8, with no modifications to the loss function. To enhance the logical flow of the article, the authors have decided not to make changes in this regard after discussion.

  1. Please add references to YOLOv5 in paragraph 1 of Section 2.4.1.

Answer: Authors really appreciate this comment. Authors apologize for not being able to cite a reference for YOLOv5, as Ultralytics, the company that open-sourced YOLOv5, has not published a peer-reviewed YOLOv5 paper. Currently, most published articles using YOLOv5 do not provide a formal citation to the original YOLOv5 paper.

  1. Please center the formula in the text.

Answer: Authors really appreciate this comment. All errors have been corrected.

  1. Please add references after CBAM and SE in the first sentence of paragraph 1 of Section 2.4.2.

Answer: Authors really appreciate this comment. The references have been added

  1. Please add references after DeepSort in the third sentence of paragraph 2 of Section 2.5.1.

Answer: Authors really appreciate this comment. The reference has been added in Sec.2.5.1

  1. Please add references after Mask RCNN and U-Net in the second paragraph of Section 2.5.2.

Answer: Authors really appreciate this comment. The references have been added.

  1. In Figure 5 and Table 1, write the model in the same order as the columns.

Answer: Authors really appreciate this comment. The error has been corrected.

  1. When drawing picture 7, the line color of the picture is not enough, which can be described with the same color + different shape patterns.

Answer: The authors appreciate the reviewer's feedback. In Figure 7(Now is F8), the selection of line colors was done thoughtfully. Data from the same photoperiod treatment groups were represented using the colors red, green, and blue, respectively, to indicate the relative strength of inter-group data. Using different colors could potentially lead to confusion among readers in distinguishing between the different photoperiod groups. Given this consideration and after thorough discussion, the authors have decided not to make any changes in this regard.

  1. The improved YOLOv8x mentioned many times in the article can be replaced with YOLOv8x+C2f-Faster-EMA.

Answer: Authors really appreciate this comment. The name "YOLOv8x+C2f-Faster-EMA" is quite long and may potentially be burdensome for readers. Therefore, after thorough discussion, the authors have decided not to make any changes in this regard. Thanks again.

  1. The advantages and contributions of the model are less described in the conclusion of the paper.

Answer: Authors really appreciate this comment. The section of conclusions have been supplemented in the paper according to the recommendations.

  1. Please change DeepSort and BotSort in the article to DeepSORT and Bot-SORT respectively.

Answer: Authors really appreciate this comment. The error has been corrected.

Reviewer 5 Report

Comments and Suggestions for Authors

1. Pconv, EMA,  BotSORT/ByteSORT/DeepSORT are well known techniques. Authors should summarize clearly what the contribution of this work is, especially in the abstract and introduction sections.

2. In Table 1, FPS should be changed to Time. In Table 2, authors need to compare the execution time, in addition to accuracy.

3. The reference list is not complete. There are many previous studies for animal tracking, and thus authors need to add the related works and compare the proposed method with the related works.

Author Response

  1. Pconv, EMA, BotSORT/ByteSORT/DeepSORT are well known techniques. Authors should summarize clearly what the contribution of this work is, especially in the abstract and introduction sections.

Answer: Authors are thankful for this comment. The main objective of this paper is to utilize algorithms for assessing the activity of meat ducks, thereby aiding in determining the impact of light cycles on meat duck activity. While the models and components used are open source, there is limited research as of now on the optimization of YOLOv8 with the C2f-EMA-Faster module. Furthermore, this study takes into account the specificity of the meat duck rearing environment when applying this technology in the meat duck farming process. Once again, we thank the reviewer.

  1. In Table 1, FPS should be changed to Time. In Table 2, authors need to compare the execution time, in addition to accuracy.

Answer: Thanks for the suggestion. The error has been corrected.

  1. The reference list is not complete. There are many previous studies for animal tracking, and thus authors need to add the related works and compare the proposed method with the related works.

Answer: Thanks for the suggestion. The paper on animal tracking has been updated. We have compared three multi-object tracking algorithms over the past two years and selected the model with the best performance. Once again, we appreciate the reviewer's feedback.

Round 2

Reviewer 1 Report

Comments and Suggestions for Authors

Dear Author(s),

The manuscript was improved after the revisions.

Best Regards.

Author Response

Dear Reviewer:
Thank you for your thorough review and valuable feedback, which have been immensely helpful for our research.
Best Regards.

Reviewer 4 Report

Comments and Suggestions for Authors

After the revision of this article, the standard of the article has been obviously improved, and the point of innovation is also very clear.

Author Response

(The authors gave the same response as above.)

Reviewer 5 Report

Comments and Suggestions for Authors

My comments were addressed except Table 2.

Please compare the execution times, in addition to MOTA, in Table 2.

Author Response

Thank you to the reviewer. Execution time information has been added in Table 2, and a comparison has been made. The comparison results have not changed the conclusion of this study. Once again, thank you for your feedback.